# Humidity Sensitivity of Chemically Synthesized ZnAl_2_O_4_/Al

**DOI:** 10.3390/s22166194

**Published:** 2022-08-18

**Authors:** Takayuki Nakane, Takashi Naka, Minako Nakayama, Tetsuo Uchikoshi

**Affiliations:** Fine Particles Engineering Group, Research Center for Functional Materials, National Institute for Materials Science, 1-2-1 Sengen, Tsukuba 305-0047, Ibaraki, Japan

**Keywords:** gas sensor, humidity, ZnAl_2_O_4_, hydrothermal synthesis, anodization, oxide, device design, environmental monitoring, semiconductor

## Abstract

Humidity sensitivity is evaluated for chemically synthesized ZnAl_2_O_4_/Al devices. We succeeded in synthesizing the ZnAl_2_O_4_/Al device by applying chemical techniques only. Hydrothermal treatment for the anodized aluminum (AlO*_x_*/Al) gives us the device of the ZnAl_2_O_4_/Al structure. All fabrication processes were conducted under 400 °C. The key was focusing on ZnAl_2_O_4_ as the sensing material instead of MgAl_2_O_4_, which is generally investigated as the humidity sensor. The evaluation of this ZnAl_2_O_4_/Al device clarified its effectiveness as a sensor. Both electrical capacitance, *C*_p_, and the resistivity, *R*_p_, measured by an LCR meter, obviously responded to the humidity with good sensitivity and appreciable repeatability. Our synthesis technique is possible in principle to improve on the process for the device with a complex structure providing a large surface area. These characteristics are believed to expand the application study of spinel aluminate devices as the sensor.

## 1. Introduction

Humidity is an important parameter. For instance, the semiconductor industry, food packaging industry, medical industry, and textile industry utilize humidity sensors for monitoring/controlling it during the production/storage process. For this usage, polymeric humidity sensors [1,2,3] have advantages for cost, but they have issues with stability, especially under conditions at high temperatures and/or at high humidity. The stability under high-humidity conditions is not only an issue for polymeric sensors but also for ceramics-based sensors. For example, humidity sensors based on aluminum oxide exhibit an excellent response [4,5,6], but high-humidity conditions have the possibility to transform aluminum oxide into boehmite (γ-AlO(OH)). Moreover, high flexibility for the shape design becomes important for all sensing devices corresponding to the advanced technology, the sophisticated industry, and the enhancement of the functionality. Standing on this background, novel ceramics sensors are still required, and numerous effective materials are also reported [7,8,9,10,11,12,13,14]. MgAl_2_O_4_ is one of the prospective materials for these discussions [15,16,17]. A wide range response (2–98 %RH) is the attractive point of this material as a humidity sensor [18]. Then, this material is chemically quite stable under atmospheres of high temperatures and high humidity. In addition, the sustainability of the resource supply and low material cost is also one of the expectable points of MgAl_2_O_4_. The most important issue for a MgAl_2_O_4_ humidity sensor is thought to be the effective technique of reducing the fabrication cost. 

The humidity sensing device of MgAl_2_O_4_ is generally fabricated as a thick film by a technique based on solid-state reactions [15,17,19,20,21]. It requires a high heating temperature of over 1000 °C. This kind of technique is reliable for study in the laboratory, but establishing the industrial fabrication technique seems to be difficult, especially when discussing the fabricating cost and the flexibility for the shape design of the product. On the other hand, thin or thick film fabrication does not require such a high temperature; however, they tend to require well-sintered powder [16,22]. These approaches also have issues with the flexibility of the shape design of the product. Standing on these considerations, the chemical synthesis route is desirable in order to develop the fabrication technique meeting the advancing requirements. Here, we inspired the chemical synthesis of MgAl_2_O_4_ from the AlO*_x_*/Al device. The chemical preparation technique of the AlO*_x_*/Al device is already established in this case as the anodization technique of aluminum, thus we should consider only the chemical reaction of MgAl_2_O_4_ from AlO*_x_*. Aluminum compounds such as spinel oxide generally require high-temperature heating conditions to form, so hydrothermal synthesis is thought to be suitable as the chemical synthesis technique for our concept. Note that hydrothermal synthesis is generally conducted at a temperature under 200 °C, but it seems to be too low to react the aluminum oxide with metal ions such as Mg^2+^. In this case, the application of high temperature over 200 °C is one approach. 

On the other hand, the verification of the target material is also meaningful. MgAl_2_O_4_ is known as typical spinel oxide with the compositional formula expressed as AB_2_O_4_, where A and B indicate the di- and trivalent metal cations, respectively. The cations existing at each site (i.e., A^2+^ and B^3+^) form the characteristic tetrahedral and octahedral structures with the coordination anion (i.e., O^2−^), respectively. The various attractive properties found in spinel oxides mainly originate in the electronic state of these two metal cations. The humidity sensitivity of MgAl_2_O_4_ is considered to originate in the connections between chemisorbed/physisorbed water and cations, even though the mechanism is not cleared from the first principal approach yet. Both MgO [23] and γ-Al_2_O_3_ with spinel structure [5] show humidity sensitivity, and this property is reported in the device of MgFe_2_O_4_ [15,24] and ZnAl_2_O_4_ [25,26]. Thus, we cannot judge easily the most important element of humidity sensitivity in MgAl_2_O_4_, but the previous report comments on the better sensitivity of MgAl_2_O_4_ than MgFe_2_O_4_ [15]. Consequently, we focused on ZnAl_2_O_4_ instead of MgAl_2_O_4_, because we experimentally felt that the fabrication of ZnAl_2_O_4_ was easier than that of MgAl_2_O_4_. In addition, ZnAl_2_O_4_ is a more famous material as a catalyst [27,28] than MgAl_2_O_4_, so ZnAl_2_O_4_ has expectable surface activity. There are few studies reporting the humidity sensitivity of ZnAl_2_O_4_, unfortunately, so identifying this property itself is thought to still be an important trial for this spinel oxide. However, the easiness of the chemical synthesis of ZnAl_2_O_4_ can become the breakthrough in the investigations of the humidity sensitivity of spinel oxide.

From this background, this work tried to fabricate the ZnAl_2_O_4_/Al device using only the water chemical technique, i.e., the anodization of aluminum and hydrothermal treatment for the AlO*_x_*/Al precursor. All of the fabrication processes were conducted under 400 °C as one of the characteristic points in contrast with numerous previous works. Then, our ZnAl_2_O_4_/Al device showed good sensitivity and appreciable repeatability.

## 2. Materials and Methods

### 2.1. Preparation of ZnAl_2_O_4_/Al Device

Figure 1 summarizes the flow of the sample preparation process with schematic illustrations of the cross-sectional image of the sample at each step and real pictures of them. The first step of the sample preparation is the anodization of an aluminum plate. The aluminum plate (99.999%, 10 × 50 × 0.5 mm) was cleaned in acetone (99.5%, FUJIFILM Wako Pure Chemical Corp., Tokyo, Japan) using the ultrasonic bath for 15 min and electropolished at 4 °C in the solution with 40 ml of perchloric acid (60%, Kanto Chemical Co., Inc., Tokyo, Japan) and 160 ml of Ethanol (99.5%, FUJIFILM Wako Pure Chemical Corp., Tokyo, Japan). The electropolish was performed with the constant current mode of 1 A for 4 min. The electropolished Al plate was washed with distilled water and dried. Then, it was anodized in the solution of 0.5 M sulfuric acid with the constant current mode of 0.01 A for 15 min at 22 °C. The solution was diluted with sulfuric acid (96.0%, Kanto Chemical Co., Inc., Tokyo, Japan) by distilled water. After washing with the distilled water and the drying processes, we could obtain the AlO*_x_*/Al precursor plate. Note that we express the amorphous aluminum oxide as AlO*_x_* in this manuscript for convenience. 

Hydrothermal (HT) synthesis was conducted for the AlO*_x_*/Al precursor plate as the next step. The precursor plate was inserted into a tube-type vessel (inner size 12 Φ× 96, Hastelloy C22, TSC-0011: Taiatsu Techno Corp., Tokyo, Japan) with 5 ml of Zn solution, and it was closed tightly. This Zn solution was created from ZnNO_3_·6H_2_O (99%: FUJIFILM Wako Pure Chemical Corp., Tokyo, Japan) and distilled water. The concentration was 0.5 M. The vessel was set in the furnace heated at 400 °C for 1 h, and it was quenched to the water bath. The inner pressure was estimated as 38 MPa, so this hydrothermal synthesis was run under the supercritical region. The obtained ZnAl_2_O_4_/Al sample was finally washed with distilled water/ethanol and dried again. 

The obtained ZnAl_2_O_4_/Al sample was masked and set in the quick coater (SC-701: Sanyu Electron Co., Ltd., Tokyo, Japan). Then, Au sputtering was conducted with the condition of 5 mA for 5 minutes (the estimated thickness of Au was 50 nm). This process formed an Au electrode on ZnAl_2_O_4_ with the size of 1 × 10 mm square (the sample discussing the humidity sensitivity at each atmosphere) or 7 mm Φ (the sample discussing the repeatability of the sensitivity). On the other hand, the sample’s edge was mechanically polished in order to remove the oxide layer for contact with the bottom Al layer.

This ZnAl_2_O_4_/Al device was glued onto the same size of PTFE plate (thickness of 1 mm) expecting mechanical support and electrical insulation. Finally, Cu-wires were connected to the top electrode and bottom Al layer at room temperature by using silver paste in an area of 1 mm Φ. Then, it was dried in air for one night at least.

### 2.2. Characterization and Evaluations

Prepared devices were characterized by a grazing incidence X-ray diffractometer (GI-XRD, Smart-Lab, Rigaku Corp., Tokyo, Japan), scanning electron microscopy (SEM, SU-8000: Hitachi High-Tech Corp., Tokyo, Japan), and energy-dispersive X-ray spectroscopy (SEM-EDS, JSM-6500F: JEOL Ltd., Tokyo, Japan) for the phase contents, the surface morphology, and the chemical homogeneity, respectively. The XRD pattern was also used for calculating the lattice constant and average grain size of the spinel layer. A large difference in the SEM images of the two different apparatus was the probe currents; they were 10 µA (SU-8000) and 104 µA (JSM-6500F), respectively. The EDS analysis requires a large probe current, although it buries the sensitive surface information.

The humidity sensitivity was evaluated by an LCR meter (IM3533: Hioki E.E. Corp., Tokyo, Japan). The schematic illustration of the measuring configuration is drawn in Figure 2. The prepared device was set in the measurement chamber (inner volume < 0.2 L) and vacuumed by a rotary vacuum pump. Here, Cu wires from the top and bottom electrodes were connected to the LCR meter of the outside in order to measure the electrical capacitance, *C*_p_, and the resistivity, *R*_p_, of the ZnAl_2_O_4_ layer. The amplitude voltage of the applied AC field was 1.0 V, and the frequency was controlled in the range between 10 Hz and 200 kHz. The monitoring pressure gauge of the chamber was a Bourdon tube type. After the equilibrium state in the chamber, an AC measurement was performed in order to obtain the calibration data as that of 0 %RH. Then, humidity-controlled gas in the gas box was introduced to the chamber. The gas box was composed of acrylic resin (inner volume 8 L), and a reference humidity meter was put inside the box (this humidity meter is a simple, commercial one. Thus, the absolute value of %RH is not thought to be sufficiently calibrated, but it was believed to be relatively reliable). After the confirmation of the gauge indicating ambient pressure, LCR measurement was performed again in order to obtain the data of each humidity. In this paper, we define *S*_C_ and *S*_R_ as the equations as follows.
*S*_C_ = (*C*_p_ (*x*) − *C*_p_ (0))/*C*_p_ (0)(1)
*S*_R_ = (*R*_p_ (*x*) − *R*_p_ (0))/*R*_p_ (0)(2)

Here, *C*_p_ (*x*) and *R*_p_ (*x*) mean the parallel capacitance and parallel resistance at the humidity of *x* %RH, respectively. Then, the *C*_p_ and *R*_p_ values in vacuumed conditions before measurement were used as the calibration data of *C*_p_ (0) and *R*_p_ (0).

The humidity in the gas box was controlled by using the saturated salt solution method. The carrier gas was ambient air. We waited until the reference sensor put in the gas box showed a stable %RH value. It was longer than 6 h at least. Table 1 summarizes the inorganic salts and %RH values used in this work. Here, ZnNO_3_·6H_2_O was exploited to build up 43 and 53 %RH. The equilibrium %RH should be a repeatable value depending on the chemical composition of the salt. However, we could obtain different stable %RH values by controlling the ratio of ZnNO_3_·6H_2_O and water. It means the quasi-equilibrium states, but the %RH value does not fluctuate after a long waiting time of over 12 h. Therefore, we used these gases with different humidity. 

## 3. Results and Discussions

### 3.1. Characterization of ZnAl_2_O_4_/Al Device

The picture of the obtained samples at each preparation step is shown in Figure 1. It demonstrates that the ZnAl_2_O_4_/Al sample (arrowed as HT) after the hydrothermal process of supercritical water for 1 h still maintains the metallic luster of aluminum substrate similarly to the samples after electropolishing and/or anodization, even though the surface slightly becomes to be cloudy. 

Figure 3 shows the XRD pattern of hydrothermally synthesized samples. The prominent peaks were all assigned as ZnAl2O4 (04-007-6610 of ICDD database) or Al (00-004-0787 of it), so that the sample is considered to consist of the ZnAl2O4 layer on the Al plate, i.e., the ZnAl2O4/Al structure as intended. The lattice constants and crystalline size of Z nAl2O4 were estimated as 8.120 Å and 5.8 nm, respectively.

Figure 4 shows the surface SEM images of the AlO*_x_* layer of the anodized AlO*_x_*/Al precursor plate and of the ZnAl_2_O_4_ layer after hydrothermal treatment. The homogeneous surface of the AlO*_x_*/Al precursor plate seems to consist of quite small grains. This morphology was speculated to become the base of the microstructure of the ZnAl_2_O_4_ layer after the hydrothermal synthesis process. However, average grain sizes seem to become small with hydrothermal treatment. The image of high magnification for the ZnAl_2_O_4_/Al sample shows the small grains with a size of less than 10 nm. This is consistent with the calculated results from XRD patterns. It implies the possibility that hydrothermal treatment promotes the sample to enhance the porous morphology.

On the other hand, the images of low magnification indicate the small grains distributed homogeneously on the surface of the sample. This homogeneity was additionally verified from the viewpoint of the element mapping for Zn, Al, and O. Figure 5 shows the EDS mapping images of the ZnAl_2_O_4_/Al sample. Some grains and different textures are slightly observed in the SEM images, although it was basically homogeneous. EDS mapping images could not detect this small roughness due to the inhomogeneity (see the images for Area 1). The relatively homogeneous distribution of each element was the same for the images of high magnification (see the images for Area 2).

### 3.2. Humidity Sensitivity of ZnAl_2_O_4_/Al Devices

The above section shows the success in the chemical preparation of ZnAl_2_O_4_/Al samples. In this section, we try to evaluate the humidity sensitivity of chemically synthesized ZnAl_2_O_4_/Al devices.

Figure 6A plots the *S*_c_ values of the ZnAl_2_O_4_/Al devices against the %RH values. The *S*_c_ value calculated by Equation (1) in Figure 6A changed corresponding to the %RH condition at the measurement atmosphere. The relationship looks linear (see blue dashed line); however, the *S*_c_ value at 0 %RH is defined as 0. Therefore, this relation is speculated to have two different slopes (see red dashed line in the figure) rather than a simple linear line (blue dashed line). This clear relationship was shown for the data measured at 10 kHz of the applied AC field. It should not be an individual relation only for the data measured at 10 kHz if the *S*_c_ values of our ZnAl_2_O_4_/Al devices truly worked as the humidity sensor. Figure 6B supports this consideration. It plots the relationship between the *S*_c_ and %RH values obtained at the different frequencies. Each relationship seems to keep the correlation that the increase in %RH enhances the value of *S*_c_ at each frequency. However, the accuracy is not thought to be constant. The sensitive response seems to be prominent corresponding to increases in the frequency; on the other hand, decreases in the values of *S*_c_ result in degrading the sensitive response. It is important for discussing the reliability of the absolute *S*_c_ value in the case of a low %RH value especially. The inserted figure in Figure 6B plots the *S*_c_ values at 52 %RH against the measurement frequency. The low-frequency data show some scattering, but it settles at the high-frequency region over 2kHz. The maximum *S*_c_ seems to be obtained at around 20 kHz. This result is not considered to be a universal trend, since the capacitance measurement using AC voltage for the insulator strongly depends on the measurement configuration. However, this result is considered to indicate at least that high-frequency measurements give us reliable values for discussing the humidity sensitivity using the capacitance data. 

On the other hand, we found an additional interesting result in the evaluating process using an LCR meter. It is shown in Figure 7 plotting the *S*_R_ values calculated by Equation (2) against the %RH values. In this case, *S*_R_ looks to change linearly corresponding to the %RH at the measurement atmosphere. The inset figure is the similar plot inserted in Figure 6B. In this case, a higher frequency measurement over 10 kHz seems to be better for obtaining reliable data.

Finally, the repeatability of the humidity sensitivity is validated for the ZnAl_2_O_4_/Al device. This experiment was conducted without a gas box since the capacity could not reproduce the same pressure for the inside of the chamber. Consequently, this property was checked by repeating the vacuuming (for setting the %RH ≈ 0) and inserting ambient air (=59 %RH on the experiment day). The total experimental time was not so long, hence we regard the humidity as constant. Figure 8 shows the time dependence of the capacitance values, i.e., *C*_p_ (0)–*C*_p_ (59) of the ZnAl_2_O_4_/Al device. Experiments were started from the vacuumed state and the leak valve was opened (gas in) by closing the valve connected to the rotary pump. Air was introduced to the measurement chamber quickly, and it was re-vacuumed again (gas out) after confirmation of the inside pressure indicating atmospheric value. This experiment was repeated three times, and the data were plotted with different colors. Figure 8A plots the data against the absolute time, and Figure 8B plots it against the relative time of each cycle. Figure 8 visibly shows the history. The response speeds seem to depend on the leaking/vacuuming rate rather than the ability of the device; however, this figure seems to demonstrate the good repeatability of the ZnAl_2_O_4_/Al. Here, dashed, short dashed, and dot lines indicate the times where the pressure gauge indicates −0.02 MPa, −0.06 MPa, and −0.08 MPa, respectively. The position of these lines also seems to replicate. The small difference is considered to be due to the low accuracy. Probably, the reproducibility of this timing would become higher if we could measure the data with the appropriate configuration recording the inner pressure accurately. The repeatability of our device was also identified for the data of *R*_p_. Figure 9 shows the results plotted in the same format as Figure 8. In this case, the data variation is wider at the higher-pressure region around the ambient atmosphere, and the start position of the gas in/gas out is clearer than that of Figure 8.

## 4. Discussion

### 4.1. Chemical Synthesis of ZnAl_2_O_4_/Al Device

This work succeeded in the chemical synthesis of the ZnAl_2_O_4_/Al device by applying hydrothermal treatment and the anodization technique. All fabrication processes were conducted under 400 °C, and it was possible in principle to improve on the process for the device with a complex structure providing a large surface area. These characteristics are believed to expand the study of the application of spinel aluminate devices as sensors. The key to this chemical synthesis is changing the target to ZnAl_2_O_4_ from the widely investigated MgAl_2_O_4_. At least, our background experiments could not synthesize a MgAl_2_O_4_/Al device by hydrothermal treatment at 400 °C using the solution formed from Mg nitrate. The XRD patterns show the Al_2_O_3_ structure. Probably, the synthesis temperature was not enough for synthesizing MgAl_2_O_4_. This trend sometimes occurs for fabricating ZnAl_2_O_4_ and MgAl_2_O_4_ by the same method. In addition, the nitrate salt used for preparing the solution of hydrothermal synthesis is also key to the success of the chemical synthesis. The reason has not been cleared yet. However, the solution of other salts drastically oxidizes and involves the device structure of AlO*_x_*/Al. Therefore, the counter anion of ZnNO_3_·6H_2_O is considered to work for preventing the metal aluminum from radical oxidation. 

On the other hand, this work prepared an AlO*_x_*/Al precursor plate created from the solution with 0.5 M of sulfuric acid. This condition is quite simple as a technique for anodizing the Al plate. For example, anodization using boric acid yields a barrier-type AlO*_x_* layer on Al; in contrast with that, sulfuric acid yields a porous-type AlO*_x_* layer [29,30]. Our experimental data shown in Figure 4 demonstrate the trend, even though it is not a generally known honeycomb structure. However, this is considered to be an important point in the future for investigating the humidity sensitivity of our ZnAl_2_O_4_/Al device. This property strongly depends on the morphology of the surface state of the spinel aluminate layer [17,19,25]. Then, Figure 4 implies that this microstructure of the final product device depends on the anodization process rather than the hydrothermal process. It means that we can chemically control the morphology of the device more intricately. 

The chemical synthesis has some advantages for the industrial application of ceramics materials; however, it also has disadvantages for fabricating the oxide, with high crystallinity exhibiting excellent performance. Chemical synthesis often degrades performance compared with that of the oxide formed by solid-state reaction. In fact, our sample is synthesized at a low temperature under 400 °C. This temperature is low enough to induce some disorders in the product oxide. The lattice constant of the ZnAl_2_O_4_ of our device seems to be longer than that of the sample sintered by a solid-state reaction at 1300 °C (≈ 8.092 Å) [31]. The reason has not been cleared yet, but cation deficiency, site exchange, impurity substitution, etc. are considered as possibilities. These considerations imply that there is further potential to improve the performance of our ZnAl_2_O_4_/Al device. This work, as the first trial for the chemical synthesis of a ZnAl_2_O_4_/Al device, has not optimized the fabricating condition of the device. Further investigation is required for optimizing it and for clarifying the correlation between the strict structural characteristics and the performance of the humidity sensor. 

### 4.2. Humidity Sensitivity of ZnAl_2_O_4_/Al Device

Experimental results for the ZnAl_2_O_4_/Al device show the relationship between the *S*_c_ and %RH values. It appears to have good sensitivity and appreciable repeatability as the humidity sensor. This point should be discussed in this section. 

At first, the relationship between *S*_c_ and %RH seems to have two different slopes (see red dashed line in Figure 6A). A previous study performed for MgAl_2_O_4_ devices reports the existence of three slopes [17]. It is a commonly discussed trend, and the difference in the dominant response is concluded as the reason for this change in the slope. Our result is also considered to be consistent with these reports. For a lower humidity region, the small slope indicates the dominance of the chemisorption water; contrary to that, the physisorption of water is dominant for the middle humidity region. Here, we can expect an additional change in the slope for the region over 70 %RH, because the condensation of water will occur in the device macrospore. However, our experiments frequently vacuumed the chamber and the porosity seemed to be low. Therefore, the influence of water condensation was considered to be small. This is the reason for the two different slopes. On the other hand, this speculation should be common with the trend of *S*_R_, although it appears to be a linear relationship. This is speculated to be due to the absolutely small value of the resistance owing to the large area of the electrode and thin thickness of the ZnAl_2_O_4_ layer.

Next, the humidity sensitivity of ZnAl_2_O_4_ should be compared with the general trend. This paper is thought to be the first report on a ZnAl_2_O_4_/Al device synthesized by a low-temperature chemical technique only. Although, previous studies report the effectiveness of ZnAl_2_O_4_ fabricated by solid-state reactions [25,26]. They report that the data originating in the capacitance value were enhanced corresponding to the increase in the %RH value. On the other hand, the data originating in the resistance value become small corresponding to the increase in the %RH value. Our experimental results shown in Figure 6 and Figure 7 are consistent with these trends. Therefore, we believe that Figure 6 and Figure 7 certainly indicate the response of the ZnAl_2_O_4_/Al device to its atmospheric humidity.

The high responsibilities shown in Figure 8 and Figure 9 can impress onto us the effectiveness of our devices as vacuum indicators. The above discussion concluded their effectiveness as humidity sensors, so this response is also considered to originate in their sensitivity to changes in atmospheric humidity. However, it is also interesting if the ZnAl_2_O_4_/Al device worked as a vacuum sensor. Figure 8 and Figure 9 show the high sensitivity of our device in a low-pressure region rather than a high-pressure one. This is considered to be the next issue for this device. In addition, ZnAl_2_O_4_ is well-known as a catalyst material [27,28]. Consequently, we speculated that a ZnAl_2_O_4_/Al sample is effective not only as a humidity sensor but also as a device for other gas detection. These points are also the next issues of our device.

Finally, both capacitance and resistance were good indicators for sensing the humidity response. We experimentally felt that the stability of the absolute measured value was better for resistance rather than capacitance. In addition, Figure 8 and Figure 9 show the high sensitivity of our device in a low-pressure region rather than a high-pressure one. This is important since a low %RH is the focus of the humidity sensor for the next generation. Standing on this viewpoint, resistivity is considered to be a good indicator. However, the inset figure in Figure 7 indicates the importance of the measurement at the high-frequency region. If we can use the device effectively at a frequency of around 60 Hz, it is expected to accelerate the application of the ZnAl_2_O_4_/Al device. This viewpoint supports the utilization of the capacitance data as the indicator.

## 5. Conclusions

This work succeeded in the chemical synthesizing of a ZnAl_2_O_4_/Al device by applying hydrothermal treatment and the anodization technique. The electrical capacitance, *C*_p_, and the resistivity, *R*_p_, measured by an LCR meter, obviously respond to changes in humidity. This clear response was obtained from high-frequency measurements.

Humidity sensitivity is basically discussed for the device of MgAl_2_O_4_, and the devices are generally fabricated by solid-state reactions applying heat treatment at higher than 1000 °C. On the other hand, the sensing material of our device was ZnAl_2_O_4_, and the preparation process was all exhibited chemically at under 400 °C. However, phase characterization indicates the device structure of ZnAl_2_O_4_/Al (see Figure 3). Then, the experimental data evaluated for this device were consistent with previous few reports about the humidity sensitivity of ZnAl_2_O_4_ [25,26]. Therefore, we concluded that our ZnAl_2_O_4_/Al device worked as a humidity sensor and that the ceramics sensor of spinel aluminate could be chemically synthesized by applying the anodization and hydrothermal technique. 

Our ZnAl_2_O_4_/Al device demonstrates good sensitivity and appreciable repeatability as a humidity sensor, although the preparation procedure was more convenient than other techniques standing on solid-state reactions or CVD or PVD. This technique is possible in principle to improve on the process for the device with a complex structure providing a large surface area. Therefore, our investigation is believed to expand the application study for spinel aluminate devices as the sensor. This work is the first trial to verify the effectiveness, thus there are some issues to investigate more carefully. For example, deep verification for the performance (reproducibility, hysteresis, response speed, life, etc.) is thought to be an issue. In addition, the sample quality, morphological analysis, device structure, and measurement configurations should be optimized appropriately. We believe that our ZnAl_2_O_4_/Al device is worth investigating for these issues. Therefore, further investigation is required.

## Figures and Tables

**Figure 1 sensors-22-06194-f001:**
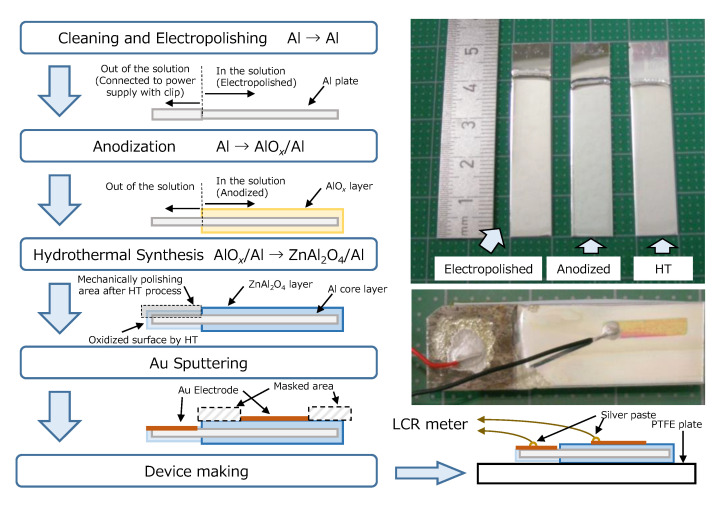
The flow of the sample preparation process with schematic illustrations of the cross-sectional image of the sample at each step and real pictures of them. The upper picture shows the scale (**left side**), electropolished Al plate (second from the left), anodized electropolished plate (AlO*_x_*/Al: second from the right) and hydrothermal synthesized sample (ZnAl_2_O_4_/Al: **right side**). Bottom picture shows the device after measurement.

**Figure 2 sensors-22-06194-f002:**
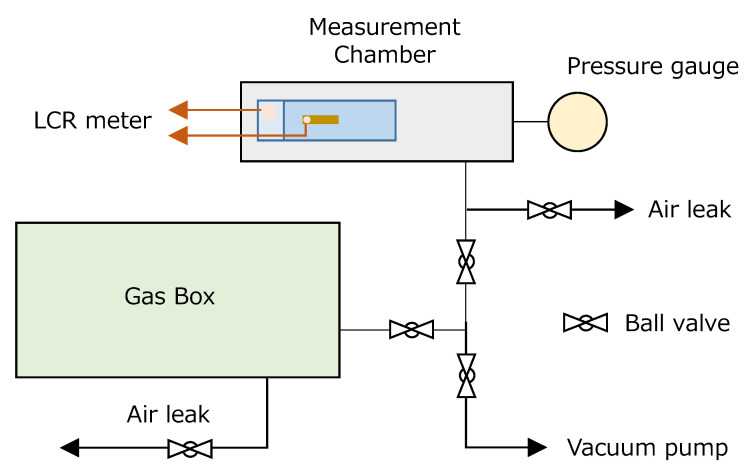
Schematic illustration of the measurement configuration for evaluating the humidity sensitivity of the sample device.

**Figure 3 sensors-22-06194-f003:**
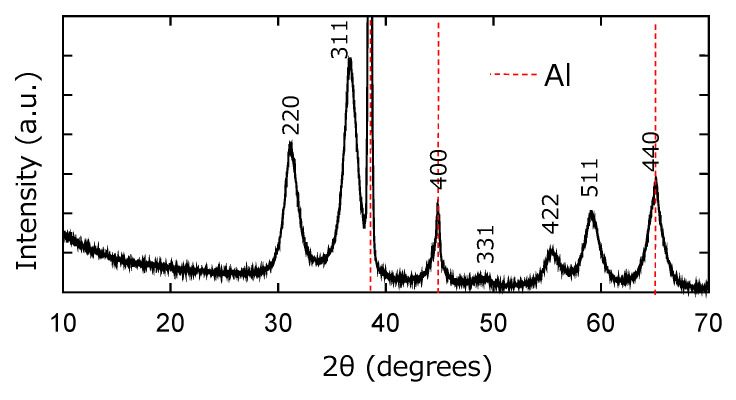
XRD patterns of the ZnAl_2_O_4_/Al sample. The peaks of ZnAl_2_O_4_ were all indexed in this figure. On the other hand, the peak positions of Al were only indicated by a red dash line. The 400 and 440 peaks were overlapped with the small Al peaks.

**Figure 4 sensors-22-06194-f004:**
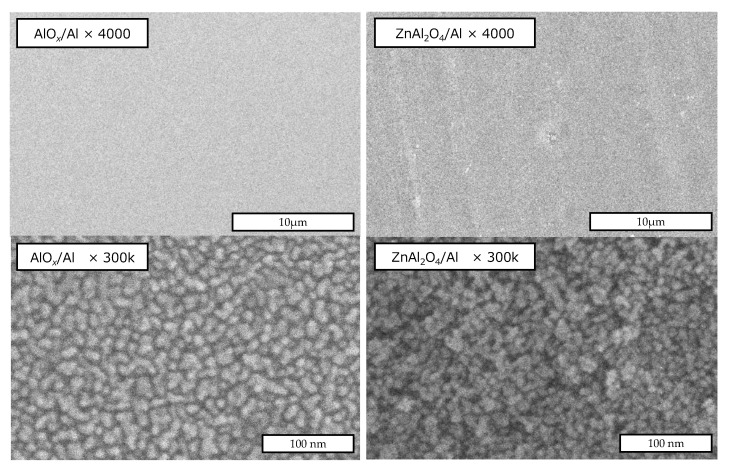
SEM images of the surface of the anodized AlO*x* layer (**left side**) and hydrothermally synthesized ZnAl_2_O_4_ layer (**right side**).

**Figure 5 sensors-22-06194-f005:**
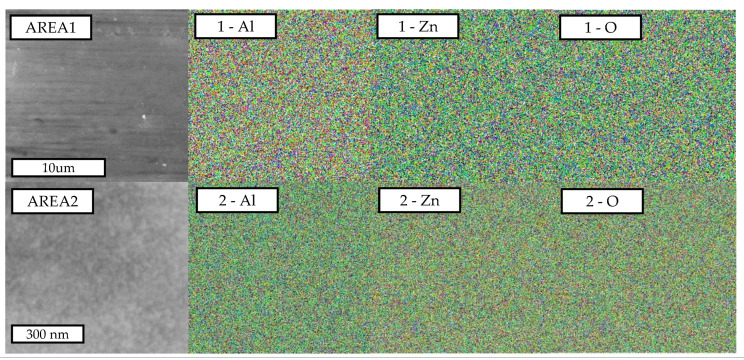
Mapping images observed by SEM-EDS for ZnAl_2_O_4_/Al sample. Area 1 shown on the top was 3000 times scale, and Area 2 (bottom) was 13,000 times, respectively.

**Figure 6 sensors-22-06194-f006:**
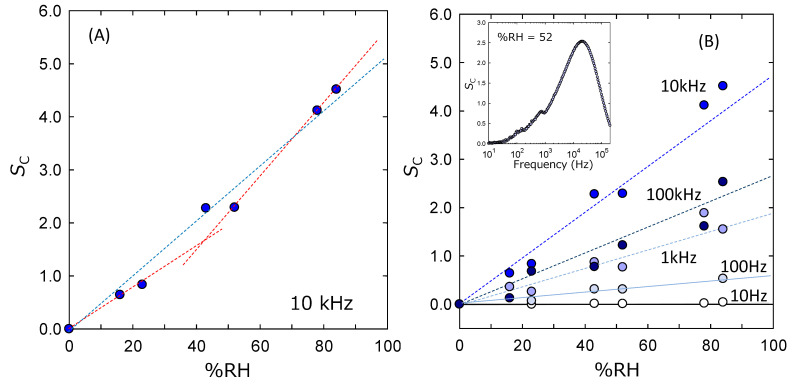
(**A**) Dependence of the *S*_c_ on the %RH in the atmosphere of ZnAl_2_O_4_/Al device. (**B**) Dependence of relationship between the *S*_c_ and %RH values on the applied AC frequency to ZnAl_2_O_4_/Al device. The inserted figure plots the *S*_c_ values at %RH = 52 against the applied AC frequency.

**Figure 7 sensors-22-06194-f007:**
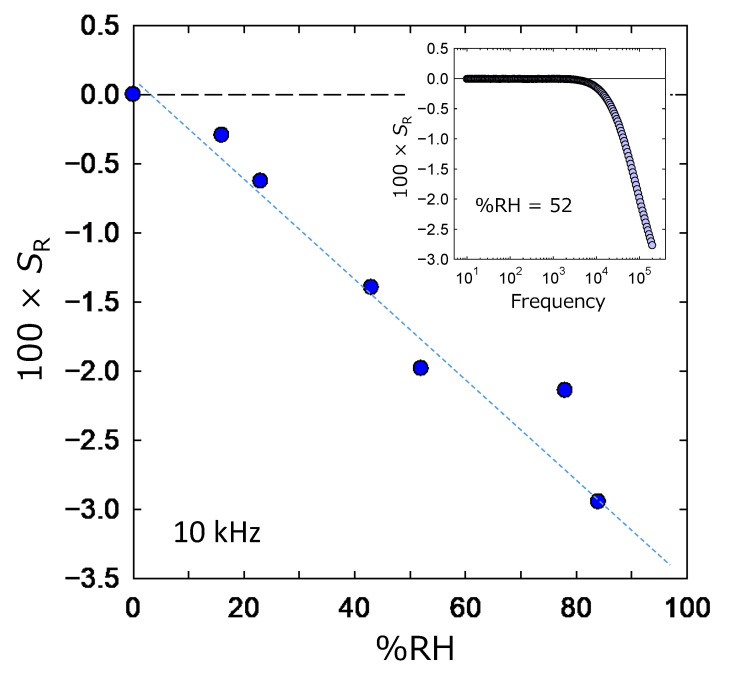
Dependence of the *S*_R_ on the %RH in the atmosphere of the ZnAl_2_O_4_/Al device. The inserted figure plots the *S*_R_ values at %RH = 52 against the applied AC frequency.

**Figure 8 sensors-22-06194-f008:**
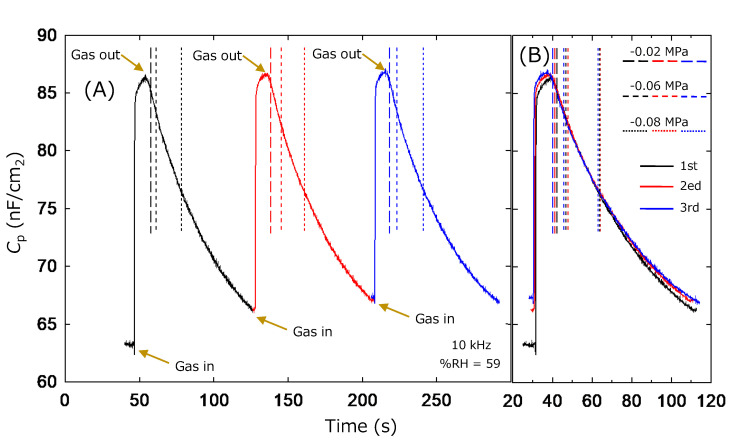
Repeatability of the *C*_p_ of ZnAl_2_O_4_/Al device against changing pressure between the vacuumed state as %RH = 0 and normal atmosphere as %RH = 59. (**A**) *C*_p_ is plotted against absolute time. (**B**) *C*_p_ is plotted against the relative time of each cycle.

**Figure 9 sensors-22-06194-f009:**
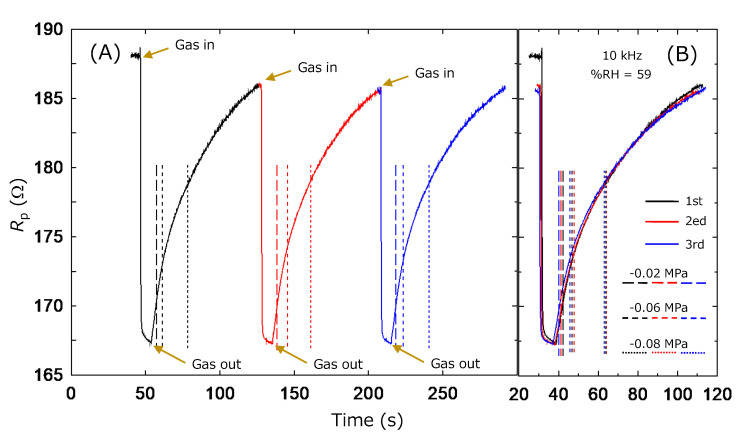
Repeatability of the *R*_p_ of ZnAl_2_O_4_/Al device against changing pressure between the vacuumed state as %RH = 0 and normal atmosphere as %RH = 59. (**A**) *R*_p_ is plotted against absolute time. (**B**) *R*_p_ is plotted against the relative time of each cycle.

**Table 1 sensors-22-06194-t001:** List of salt used for controlling the %RH value in the gas box.

Salt	%RH
MgCl_2_	16
CH_3_COOK	23
ZnNO_3_·6H_2_O	43
ZnNO_3_·6H_2_O	52
NaHCO_3_	78
Na_2_SO_4_	84

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
