# Peer review of "Humidity Sensitivity of Chemically Synthesized ZnAl2O4/Al"

_sensors, 2022, doi:10.3390/s22166194_

Round 1

Reviewer 1 Report

The authors present a humidity sensor based on a ceramic material. Per se, it is not very exciting and the authors justified their work in a good introduction. The paper is well structured but needs a lot of improvements. My remarks are listed below:

1. Add a schematic flow sheet of the synthesis steps.

2. Add a picture of the final sensors

3. line 111. Sentence is not clear. Does it mean the aluminum substrate is polished away ? If it is the case, it is a difficult step in the preparation of the sensor and we should consider a chemical etching instead.

4. line 137. Sentence about accuracy is not clear.

5. peaks 400 and 440 assigned to Al are rather broad. Why ?

6. The authors should give an equivalent electrical circuit for their sensor. Show the Nyquist and Bode representations of the impedance.

7. Insets in figure 6 and 7 have no label on vertical axis.

8. What about the contact resistance between gold electrodes and ceramic sensitive layer?

9. Fig 9 should be put in section 3.

10. The experimental set up does not allow to conclude on response and recovery times.

11. The effect of humidity depends on the adsorption isotherms which are usually not linear and in some cases, present a hysteresis. Can you comment on that?

12. Resistance change seems rather small and the resistance seems quite small which is strange for a so-called insulating ceramic? It is due to the short circuit by remaining Al or to the conductivity of the ceramic itself?

Reviewer 2 Report

This manuscript reported a low-temperature synthetic method of ZnAl2O4/Al with high humidity sensitivity in resistance and capacitance. The authors provided sufficient experiments and discussions, so that I recommend it to be published in Sensors after a revision addressed a few comments.

1.      The PDF Card No. of ZnAl2O4 should be provided.

2.      What is the carrier gas of humid gas? Air or pure N2?

3.      The calibration data of Cp(0) and Rp(0) are tested in vacuum. In comparison, how are the Cp and Rp in dry air or pure N2?

4.      The English should be improved. To many grammar errors such as,

Line 122, “The XRD pattern was also used for calculate the lattice constant……”

Line 181, “It implies the possibility that hydrothermal treatment promote the……”

Line 237, “This experiments was examined without gas box……”

Line 242, “Experiments were start from vacuumed state……”

Line 247, “…… and (b) plot it with the relative time of each cycle.”
